# Exploring emotion recognition in patients with mild cognitive impairment and Alzheimer's dementia undergoing a rehabilitation program

**Masaki Kamiya**[1]*, **Aiko Osawa**[1], **Eri Otaka**[2], **Kenji Kato**[3], **Tatsuya Yoshimi**[3], **Hitoshi Kagaya**[1], **Izumi Kondo**[4]

**1** Department of Rehabilitation, National Center for Geriatrics and Gerontology, Obu, Aichi, Japan, **2** Laboratory of Practical Technology in Community, Assistive Robot Center, National Center for Geriatrics and Gerontology Research Institute, Obu, Aichi, Japan, **3** Laboratory of Clinical Evaluation with Robotics, Assistive Robot Center, National Center for Geriatrics and Gerontology Research Institute, Obu, Aichi, Japan, **4** National Center for Geriatrics and Gerontology, Obu, Aichi, Japan

* mkamiya@ncgg.go.jp

## Abstract

### Aim

This study aimed to explore differences in the emotions of patients with mild cognitive impairment (MCI) and Alzheimer's dementia (AD) in group rehabilitation using facial analysis.

### Method

We conducted rehabilitation consisting of aerobic exercise, cognitive training, dual tasks (a combination of exercise and cognitive training), and creative activities in a group format with patients with MCI and dementia. The faces of the 30 patients (MCI: n=14; mild/moderate AD: n=16) who participated were filmed from the front with a small camera during the four tasks. Then, we used the Kokoro Sensor (CAC Corporation, Japan), a device which estimates emotion scores (anger, contempt, disgust, fear, joy, sadness, surprise) based on different parts of the face using artificial intelligence, to calculate emotion scores for each activity, and compared them between the MCI and AD groups.

### Results

Emotion scores for fear and surprise were significantly higher for the AD group than for the MCI group during dual tasks (p=0.016), while emotion scores for joy were significantly higher for the MCI group than for the AD group during creative activities (p=0.012).

**Data availability statement:** All relevant data are within the paper and its Supporting Information files.

**Funding:** 1. Japan Society for the Promotion of Science Grants-in-Aid for Scientific Research Grant Number JP21K11209 2. Research Funding for Longevity Sciences (24-24) from the NCGG The funders had no role in study design, data collection and analysis, decision to publish, or preparation of the manuscript.

**Competing interests:** The authors have declared that no competing interests exist.

## Conclusion

Creative activities and dual tasks, which require simultaneous physical activity and cognitive thinking, were difficult for patients with AD. On the other hand, tasks which used a range of cognitive functions, such as creative activities, evoked joy in patients with MCI. It may be beneficial to provide tasks and support to patients with respect to their unique emotions based on these results.

## Introduction

As the prevalence of dementia continues to rise in Japan, institutions such as care facilities for the elderly and hospitals have begun to provide group activities for the purposes of maintaining physical function and increasing opportunities for activity for people with dementia. Group activities are defined as activity programs focused on plentiful sensory and motor stimulation, such as exercise and creative pursuits, conducted in a group setting [1]. Small group activities in particular are reported to have higher participation rates [2]. One report has also indicated that group activities involving robots for individuals with dementia can reduce nighttime awakenings and improve sleep quality [3]. Additionally, family members and medical professionals managing individuals with dementia have indicated that group activities are suitable for this population, provided that programming remains flexible [4]. Nonetheless, there is still inadequate evidence to support the effects of group activities on people with dementia [5,6]. Further, dementia rehabilitation should seek to actively stimulate brain activity and should be performed using errorless learning [7]. The content of the activities offered should also be individualized in accordance with the severity of the decreased cognitive function and motivation caused by dementia so that it is at an appropriate difficulty level and can maintain enthusiastic participation [8]. In other words, it is essential to think about rehabilitation programs in terms of the respective abilities of the patients, for example the difference in severity between mild cognitive impairment (MCI) and Alzheimer's dementia (AD) (Fig 1).

However, deciding what kind of activities are appropriate for people with MCI or AD is no easy task. One way to do so is to evaluate the facial expressions of people with dementia in response to their own behaviors or the care received by medical staff [9,10]. While it is important to consider whether the care performed is appropriate for the patient, there are few evaluations that objectively measure whether a treatment with appropriate content and the correct difficulty level was provided. Moreover, the reliability of patients' answers has been called into question considering the declines in verbal expression and comprehension that exist in dementia [11]. Thus, accurate evaluation requires an objective method such as facial analysis.

Still, facial analysis is based on observation, and poses a challenge in that it requires manpower and expertise. To resolve this challenge, recent years have seen a growing number of studies of facial analysis using artificial intelligence (AI) based on facial expression recognition [12–14]. In this method, the Facial Action Coding System (FACS), which codifies combinations of the movements of various

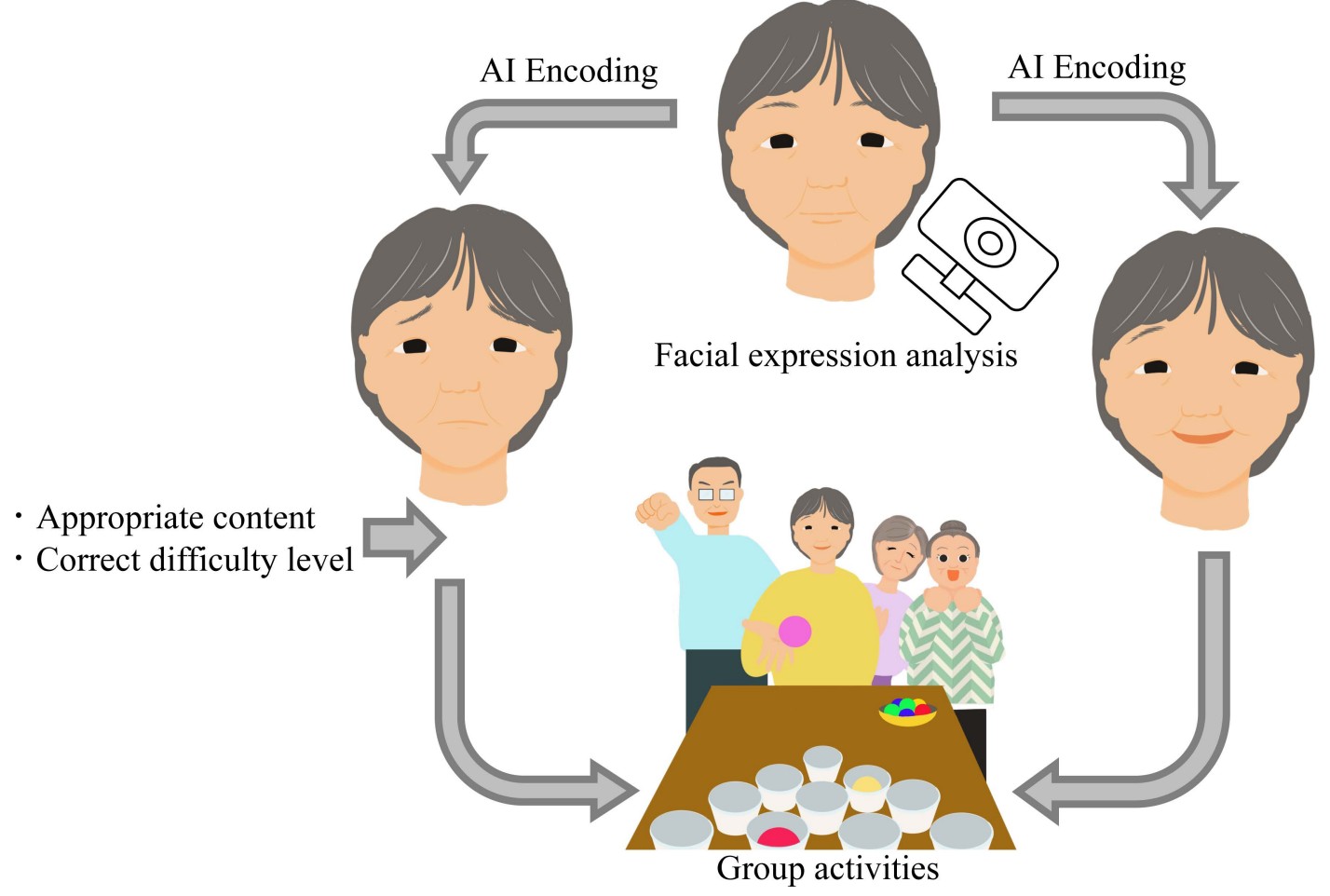

**Fig 1. This is the preference for group activities for people with MCI and dementia using AI-based facial expression analysis.**

facial muscles into discernible patterns [15], is used to perform automatic facial expression analysis (AFEA), a method of automatically converting the position of facial muscles into basic, universal emotions using artificial intelligence. For example, Garrido et al. (2018) captured changes in facial expression using facial electromyography to investigate music preferences among patients with dementia, and it is likely that facial analysis using AI could capture changes in facial expression in the same way [16]. Saposnik et al. (2019) previously employed AFFDEX to explore the relationship between emotions, affective states (as interpreted through facial muscle activity and emotional expressions), and therapeutic inertia in physicians' treatment decisions for patients with multiple sclerosis [17]. Their findings revealed that facial expressions (e.g., frowns, nose wrinkles) and emotional expressions (e.g., disgust) influenced doctors' decisions, with aversion to ambiguity partially mediating therapeutic inertia. Additionally, prior studies have reported favorable inter-reliability outcomes when assessing pain in dementia patients using AI-based facial expression analysis [18]. However, no research utilizing facial expression analysis to optimize group activities for dementia patients has yet been conducted.

Accordingly, in this study, we attempted emotion analysis of the facial expressions of patients with MCI and AD during group rehabilitation using AI with the aim of exploring differences in emotion between MCI and AD using facial analysis. We also used this emotion analysis to investigate the participation level, task preferences, and directions for support in the context of group activities for people with MCI and dementia.

## Materials and methods

### Study participants

Participants were 30 individuals who were seen at the National Center for Geriatrics and Gerontology (NCGG) and underwent outpatient rehabilitation between December 21 2021 and July 30 2023. Participants were provided informed consent in writing and written consent was obtained. If the participant was unable to understand the content, consent was obtained from the family caregiver.

These individuals were diagnosed with MCI (n=14) or AD (n=16). Diagnosis of MCI followed Petersen's criteria, and all cases were of amnestic MCI. AD was diagnosed based on the National Institute of Neurological and Communicative Disorders and Stroke/Alzheimer's Disease and Related Disorders Association (NINCDS-ADRDA) Alzheimer's Criteria, and cases diagnosed with probable AD, possible AD, or AD + cardiovascular disease (CVD) were included. Patients with dementia other than AD or MCI, for example dementia with Lewy bodies (DLB), vascular dementia (VaD), or frontotemporal dementia (FTD), or a history of neurological/psychiatric disease (for example, depression) or other disease (for example, alcohol use disorder) were excluded from the study.

### Rehabilitation

The dementia rehabilitation offered at the NCGG is a comprehensive rehabilitation program for outpatients and their family caregivers. The program meets once a week for 60–80 minutes with a class size of 4–8 participants, and is personalized with one therapist per family. The program consists of four types of activities: a. aerobic exercise, b. cognitive training, c. dual task training combining therapeutic exercise and cognitive training, and d. creative activities. Specifically, aerobic exercise lasts for approximately 20 minutes and consists of stomping and stepping movements in a standing position while making simple arm movements as directed by an instructor. Cognitive training is done in a seated position and focuses mainly on an attention task of quickly matching playing cards by color or suit, and the memory task constitutes memorizing numbers or pictures using picture cards and recalling them some minutes later. Dual tasks consist of stomping or stepping to the rhythm in a standing position while simultaneously answering cognitive tasks projected on a screen at the front of the room. Creative activities consist of handicrafts in which patients make a piece of art with their hands using things such as paper, coloring materials, tree pieces, glue, and scissors. These rehabilitation sessions are provided by occupational therapists, physical therapists, and speech therapists based on the instructions of rehabilitation physicians and the features of dementia.

### Assessment

The primary outcome was participants' emotion scores calculated by facial analysis from the recordings taken during rehabilitation. The secondary outcome was participants' feelings of fun as measured by visual analog scale (VAS) [19] for each task after the rehabilitation session was completed. The VAS ranged from 0 (extremely boring) to 10 (extremely fun). A reaction scale [20] was also used to evaluate autonomy and participation during rehabilitation sessions. In addition, basic attributes of age, sex, education history, and disease duration were collected. For evaluation of mental function, depression was evaluated using the Self-rating Depression Scale (SDS) [21], and subjective health was evaluated using the General Health Questionnaire (GHQ) [22]. Depression is considered a risk factor or precursor for dementia [23], while its presence should be assessed as it may influence both facial expressions and subjective well-being. For evaluation of cognitive function, the Mini-Mental State Examination-Japanese (MMSE-J) [24] was used. We further used the Frontal Assessment Battery (FAB) [25], which is associated with dementia assessment scales and can be used to assess frontal lobe dysfunction, as well as Raven's Progressive Matrices (RCPM) [26], which evaluates visuospatial cognitive function. The Neuropsychiatric Inventory (NPI) [27] was used to evaluate behavioral and psychological symptoms of dementia (BPSD). Further, in order to indicate the daily activity level, activities of daily living (ADL) were evaluated using the Barthel Index (BI) [28] and the Frenchay Activities Index (FAI) [29] (Fig 2).

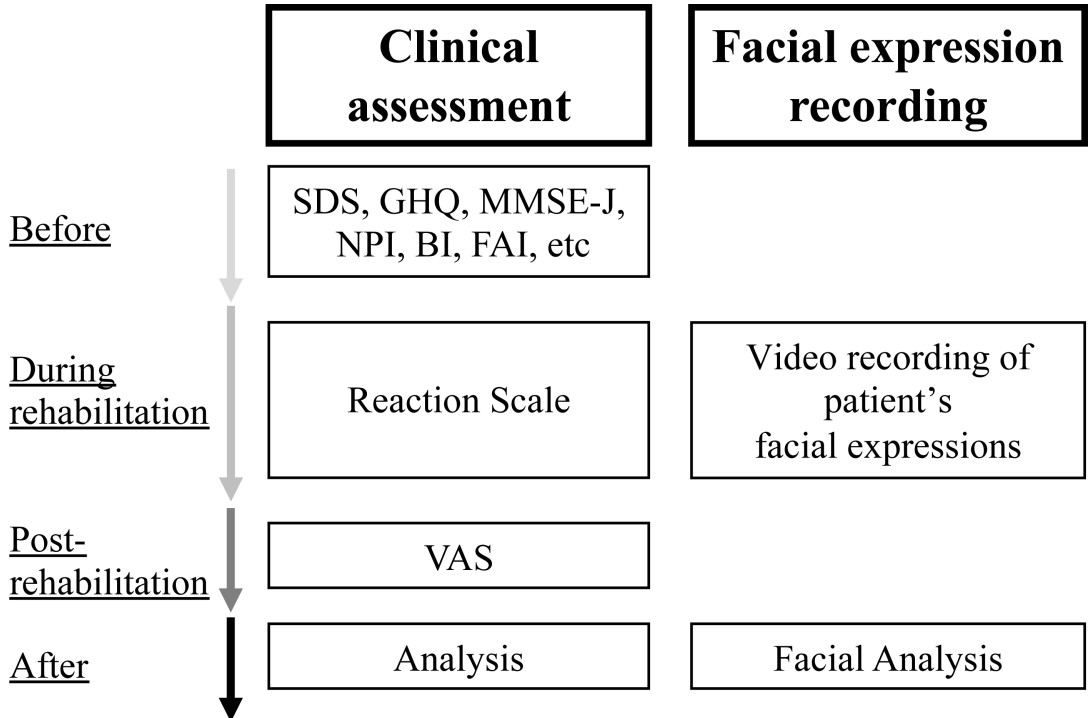

**Fig 2. This is the clinical assessment facial expression recording process.** SDS: Self-rating Depression Scale, GHQ: General Health Questionnaire, MMSE-J: Mini-Mental State Examination Japanese, NPI: Neuropsychiatric inventory, BI: Barthel Index, FAI: Frenchay Activities Index, VAS: Visual Analog Scale.

## Procedure

Patients' facial expressions were recorded using the Insta360 GO 2 (Insta 360 Japan Co., Ltd.). Similar to the Insta360 GO 3 (Insta 360 Japan Co., Ltd.), the GO 2 is the world's smallest action camera and was chosen to minimize distraction for participants during rehabilitation. For programming conducted in a standing position (a. aerobic exercise and c. dual task training), the camera was positioned approximately 1 m in front of the participant using a tripod and the height adjusted to that of the participant's face. For programming conducted while seated (b. cognitive training and d. creative activities), the camera was set up on a desk to face the participant.

Facial analysis of the captured video used Kokoro Sensor (CAC Corporation, Tokyo, Japan). The calculation of emotion scores with Kokoro Sensor is based on Ekman's (1978) FACS theory [15], which has undergone training using over 9 million data points. With this system, approximately 7 billion data points on the face are divided into eye movements and mouth movements and analyzed to estimate seven emotion scores (anger, contempt, disgust, fear, joy, sadness, and surprise) and neutrality on a scale from 0 to 100. The validity of the analyzed scores and facial muscle movements has been verified [30]. As the recordings for this study were taken during the Coronavirus disease 2019 (COVID-19) pandemic, participants wore transparent, acrylic masks over their mouths to prevent the spread of infection.

## Analysis

First, mean ± standard deviation (SD), median (interquartile range: IQR), and total number (%) were calculated for each outcome for patients with MCI and AD, and participants' basic attributes were compared. Based on the distribution normality, statistical analyses used either the unpaired t-test, Mann Whitney U-test, or chi-squared test.

Second, the analyzable percentage was calculated after analysis with Kokoro Sensor to investigate the feasibility of facial analysis during group rehabilitation. This was calculated using the number of frames with analyzed values out of the total number of frames for all tasks. The analysis interval was from the point at the start of each task when the participant either began to move their body or picked up the items for the task, to the point at which the participant was told by the therapist that the task was over and their hands stopped moving.

Third, we investigated the differences between MCI and AD for each emotion score, the VAS, and the reaction scale for each of the four rehabilitation tasks. Here, the average scores for each emotion over the duration of the analysis interval were used as emotion scores. As this study was evaluating participants during normal rehabilitation, there were conversations, breaks, and time spent waiting. Therefore, moments with a neutral score of ≥50 were not used; only moments with a neutral score of <50 were included in the analysis of emotion scores. Based on the distribution normality, statistical analyses used either the unpaired t-test or Mann-Whitney U-test.

Fourth, to look at correlations between joy emotion score calculated by Kokoro Sensor, the VAS, and the reaction scale, either the Pearson product-moment correlation coefficient or Spearman's rank correlation coefficient was calculated depending on the distribution normality.

All analyses were performed in SPSS, version 28.0 (IBM Corporation, Armonk, NY, USA), with a significance level set at p<0.05.

### Ethics statement

This was a cross-sectional study and involving humans were approved by the Ethical Review Board of Japan's National Center for Geriatrics and Gerontology (NCGG) (No. 1553). The studies were conducted in accordance with the local legislation and institutional requirements. The participants provided their written informed consent to participate in this study. The individuals pictured in S1 Fig and S2 Fig have provided written informed consent (as outlined in PLOS consent form) to publish their image alongside the manuscript.

## Results

### Demographics

MMSE-J (p<0.001), RCPM (p=0.001), NPI (p=0.010), and GHQ (p=0.045) differed between the MCI and AD groups. Age, sex, education history, disease duration, FAB, BI, FAI, and SDS did not differ significantly between the two groups (Table 1).

### Emotion score analyzable percentage

Recording time was 13.96 ± 4.78 min for aerobic exercise, 15.68 ± 4.11 min for cognitive training, 10.62 ± 6.44 min for dual tasks, and 19.51 ± 5.12 min for creative activities. The percentage analyzed was 46.99 ± 28.16% for aerobic exercise, 54.33 ± 37.41% for cognitive training, 54.16 ± 34.01% for dual tasks, and 46.55 ± 39.88% for creative activities.

### Comparison of MCI and AD emotion scores, VAS, and the reaction scale during rehabilitation

For emotion scores, fear and surprise scores were significantly higher in the AD group than in the MCI group during dual tasks (p=0.016), while joy score was significantly higher in the MCI group than in the AD group during creative activities (p=0.012) (Fig 3). No significant differences were observed between the two groups for other tasks or emotion scores.

No significant difference was observed between the MCI and AD groups for the VAS or the reaction scale, with both generally rating the activities as fun and task participation as enthusiastic (Table 2).

### Correlation between emotion score and VAS

There was no significant correlation between joy emotion score and VAS in either group for all tasks (Fig 4.).

**Table 1. Comparison of the MCI and AD.**

| | MCI (n=14) | | AD (n=16) | | p-values |
|---|---|---|---|---|---|
| Age (years) † | 77 | (6) | 80 | (7) | 0.276 |
| Male (%) ‡ | 8 | (57.1) | 7 | (43.8) | 0.464 |
| Education (years) † | 13 | (2) | 12 | (2) | 0.096 |
| Estimated onset of dementia (months) † | 70 | (30) | 88 | (44) | 0.240 |
| MMSE-J § | 24 | [24–25] | 19.5 | [13–21] | **<0.001** |
| FAB § | 14 | [12–15] | 10 | [9–13] | 0.059 |
| RCPM § | 32 | [28.5-34] | 24 | [19-25.5] | **0.001** |
| NPI § | 0 | [0-2] | 3 | [1–8] | **0.010** |
| BI § | 100 | [100-100] | 100 | [100-100] | 0.648 |
| FAI § | 23 | [18–27] | 16.5 | [11-22.5] | 0.114 |
| SDS § | 35 | [30–40] | 32 | [27.5-35.5] | 0.983 |
| GHQ § | 0 | [0-1] | 2 | [1-4.5] | **0.045** |

The average (standard deviation: SD), total number (%), and median [interquartile range: IQR] of each MCI and AD endpoint were calculated.

†Unpaired t-test,

‡Chi-squared test,

§Mann-Whitney U test.

MCI: mild cognitive impairment, AD: Alzheimer's disease, MMSE-J: Mini-Mental State Examination Japanese, FAB: Frontal Assessment Battery, RCPM: Raven's Colored Progressive Matrices, NPI: Neuropsychiatric inventory, BI: Barthel Index, FAI: Frenchay Activities Index, SDS: Self-rating Depression Scale, GHQ: General Health Questionnaire.

## Discussion

### Feasibility of facial analysis during rehabilitation amid COVID-19 countermeasures

Previous studies of facial analysis have stated that an analyzable percentage of less than 20% is low [31,32]. In this study, analyzable percentages were particularly high for the cognitive and dual tasks in which the target objects or task pre-senter were observed head-on. However, facial analysis was still difficult at approximately 55% of the time. This might be because of the difficulty in capturing facial expressions due to body movement during aerobic exercise or because participants tended to focus on the detail-work of their creative projects, resulting in an angle that was outside the analyzable range of the Kokoro Sensor. In addition, because this was a group rehabilitation setting, participants wore clear masks to prevent the spread of infection during the COVID-19 crisis, but interference with the facial data points due to the mask frame or reflected light might have reduced the analyzable percentage. Nonetheless, using strategies such as modifying the camera positioning, we were able to achieve facial analysis of at least a 45% portion for each of the four tasks measured, thereby ensuring a high standard compared to past studies.

### Differences in MCI and AD emotion scores

In the dual task, emotion scores for fear and surprise were significantly higher in AD than in MCI. Due to the severe impairment of general executive resources in AD, cognitive tasks such as carrying on a conversation can impact walking speed and motor precision [33]. In addition, older adults who easily exhibit dual task interference have a high fall risk [34]. With AD in particular, there is an association between physical abilities/psychiatric symptoms and fear of falls, and this fear may become more severe as dementia progresses [35]. Furthermore, events that are unexpected or run counter to one's expectations evoke surprise [36]. The cognitive task aspect of a dual task may cause especially high levels of psychological stress in patients with AD who have reduced cognitive function. For these reasons, it can be concluded

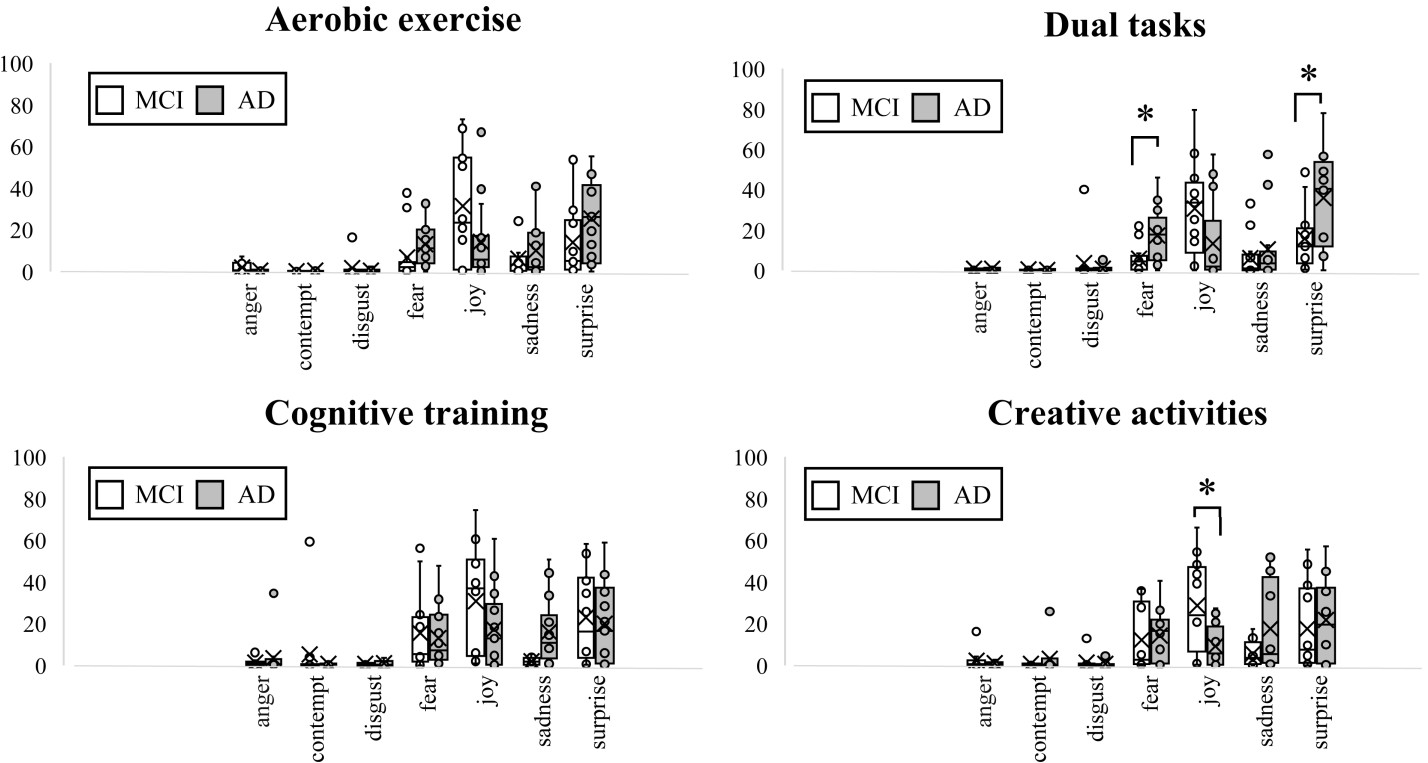

**Fig 3. This is the differences in each emotion during rehabilitation.** The average (standard deviation: SD) of each MCI and AD endpoint were calculated. Unpaired t-test. * Significant at the 0.05 level. MCI: mild cognitive impairment, AD: Alzheimer's disease.

**Table 2. Comparison of the response to rehabilitation between MCI and AD.**

| | Aerobic exercise | | Cognitive training | | Dual tasks | | Creative activities | | p-values |
|---|---|---|---|---|---|---|---|---|---|
| | MCI | AD | MCI | AD | MCI | AD | MCI | AD | |
| VAS† | 8.0(1.8) | 8.3(1.6) | 7.1(1.7) | 8.7(1.7) | 7.9(1.9) | 8.4(1.6) | 8.6(1.0) | 8.5(1.4) | 0.624ª, 0.050ᵇ, 0.475ᶜ, 0.782ᵈ |
| Reaction Scale‡ | | | | | | | | | |
| Degree of participation | 4[4–4] | 4[4–4] | 4[4–4] | 4[4–4] | 4[4–4] | 4[4–4] | 4[4–4] | 4[4–4] | 1ª, 0.350ᵇ, 1ᶜ, 0.350ᵈ |
| Autonomy | 4[4–4] | 4[4–4] | 4[4–4] | 4[4–4] | 4[4–4] | 4[4–4] | 4[4–4] | 4[4–4] | 1ª, 0.350ᵇ, 1ᶜ, 0.350ᵈ |

The average (standard deviation: SD), and median [interquartile range: IQR] of each MCI and AD endpoint were calculated.

[a]Aerobic exercise MCI vs AD,

[b]Cognitive training MCI vs AD,

[c]Dual tasks MCI vs AD,

[d]Creative activities MCI vs AD

†Unpaired t-test,

‡Mann-Whitney U test.

MCI: mild cognitive impairment, AD: Alzheimer's disease, VAS: Visual Analog Scale.

that emotions of anxiety about falling and surprise due to stress were higher in the AD group during the dual task training. These results suggest the need to provide patients with AD with support, particularly for psychological stress and fear of falling, when having them simultaneously perform a motor and a cognitive task.

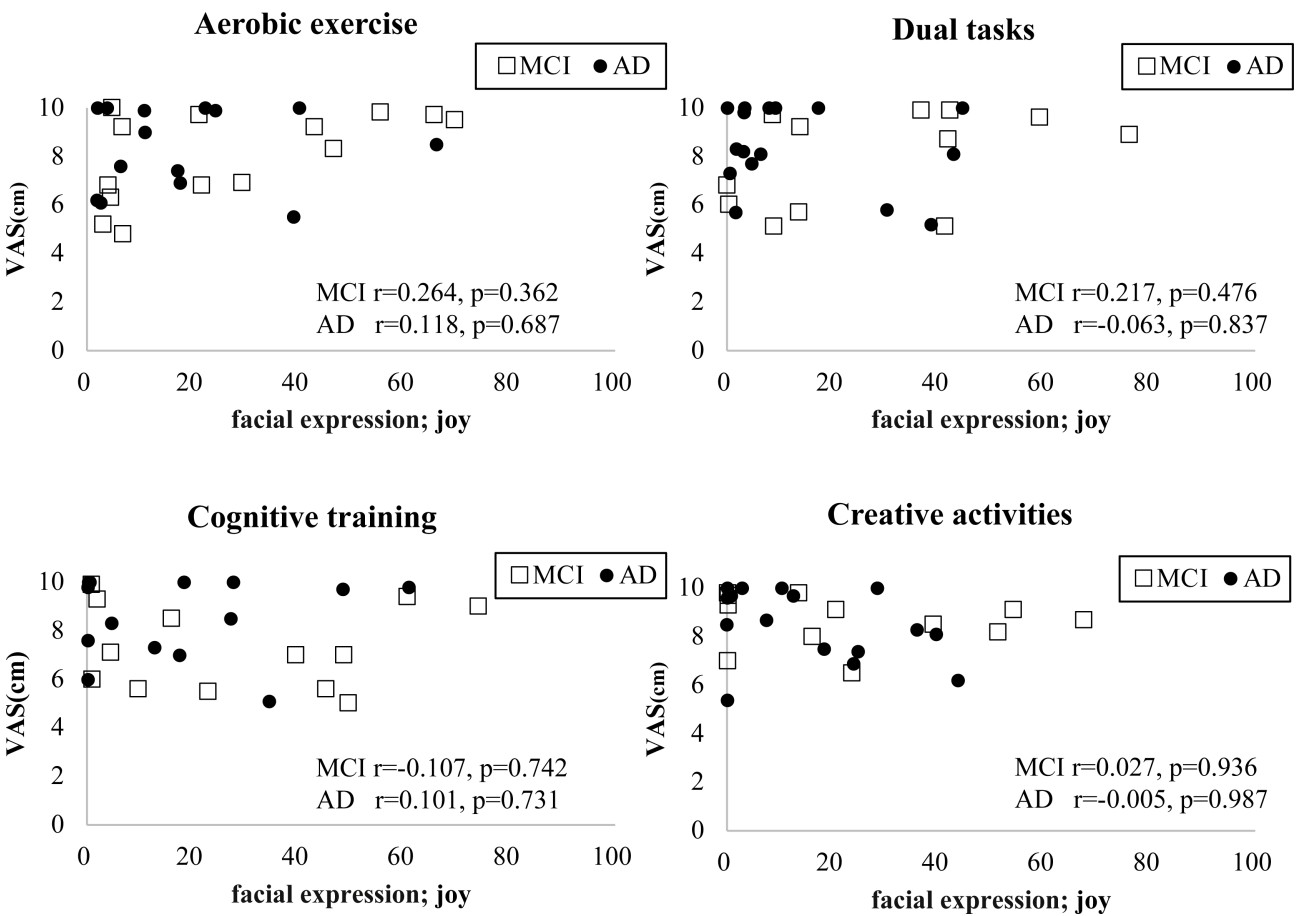

**Fig 4. This is the relationship between facial expression and VAS.** Pearson product-moment correlation coefficient. VAS: Visual Analog Scale, MCI: mild cognitive impairment, AD: Alzheimer's disease.

For the creative task, emotion score for joy was significantly higher in MCI than in AD. A report on visuospatial cognitive function in AD found lower accuracy and longer response time for the mental rotation task [37]. Based on participant demographics, the AD group had significantly lower MMSE-J and RCPM scores than the MCI group, and there were clear declines in general cognitive and visuospatial cognitive functions in AD. Thus, creative tasks that require visuospatial cognitive function are likely to be highly difficult for patients with AD, suggesting the need for support that compensates for declines in function or strategies to make the process of creating easier. On the other hand, patients with MCI were able to easily gain a sense of achievement from tasks in which multiple cognitive functions were used to complete a piece of art, suggesting that it would be effective to actively incorporate these tasks.

Lastly, contrary to the facial analysis results, there were no significant differences between the VAS and reaction scale scores for the four tasks. As such, the preferences for each task differ between patients with MCI and AD due to variations in cognitive function. Therefore, when presenting tasks to patients with MCI or AD, medical professionals can enhance motivation for continued prevention and treatment by accurately capturing each patient's facial expressions, and adjusting the tasks accordingly. The VAS showed very similar overall scores which raises questions about the reliability of the evaluation similar to those in a previous study by Scherder et al. (2000) [11]. For the reaction scale, observation results identified enthusiastic participation for every task. However, one limitation of this

observational evaluation is that it cannot predict the emotion with which the individual is engaging with the task, which demonstrates the importance of observational evaluations such as facial analysis that can better grasp the target's internal state.

## The relationship between facial analysis emotion scores and patients' subjective evaluations

There was no significant correlation between joy score from facial analysis and the VAS score. This may be, as mentioned previously, due to a problem of the reliability of responses from people with impaired cognitive function. Moreover, the evaluation timing also differed. Facial analysis is capable of calculating timely and objective emotion scores during rehabilitation, while the VAS represents the patient's subjective mood after the completion of rehabilitation. For patients with MCI and AD, who have impaired recent memory, it may have been difficult to reflect emotions from the time of the task on the VAS. While this research is from a different field, one study of pain among dementia patients proposed evaluating persistent pain based on facial expression and demeanor to understand pain objectively, rather than relying on self-report, because it is difficult for dementia patients to accurately express their pain [38]. As this proposal suggests, if we consider that patients with MCI and dementia have reduced ability to look back and express their emotions later, it is important to perform evaluation in real time using tools like facial analysis and measurement of biological responses. That being said, a review study comparing subjective and objective evaluations of patients' treatment courses concluded that, while objective evaluation is useful in identifying the underlying cause of pathology, subjective outcome is influenced by patient priorities and lifestyle; therefore, it is also essential to consider subjective evaluations in a standardized way [39]. Thus, it can be concluded that considering both objective, real-time analysis and patients' subjective evaluations will provide a more accurate understanding of a patient's emotions in clinical settings. In particular, people with dementia may struggle to express their feelings appropriately, due to factors such as reduced spontaneity and language disorders, and may refuse medical treatment or care, making it difficult to provide adequate care. It has also been reported that one in two people with dementia do not voluntarily seek medical treatment [40]. By objectively evaluating facial expressions during interventions, along with asking the patient about the situation afterward, it may be possible to more accurately understand each patient's emotions. This could enable the provision of medical care that respects the individuality of each patient with dementia, such as adjusting the difficulty of interventions and selecting activities that interest them.

## Limitations

Due to the ongoing COVID-19 infection countermeasures, facial expressions were recorded while wearing transparent, acrylic masks. A pilot study found no difference in the rough numbers of whether a transparent mask was worn; however, it is possible that facial data points may have been concealed by reflections off the transparent mask. Further, there are two types of smiles: a "felt-smile" and an "unfelt-smile" [41]. The facial movement of a felt-smile is characterized by a lifting of the lower eyelid, narrowing of the eyes, the appearance of crow's feet wrinkles at the corners of the eyes, and a tight pulling back of the corners of the mouth [42]. On the other hand, in an unfelt-smile, there is no movement of the orbicularis oculi muscle; it is a smile with the corners of the mouth only. Humans are able to distinguish and recognize these two smiles [43,44], but the Kokoro Sensor does not detect the two distinctly. Thus, the analysis might have underestimated smiles (in other words, joy) among the patients with dementia who tend to have fewer facial expressions. This is a technical issue that should be addressed in the future. In this study, no established reference values exist for interpreting the emotional scores calculated by Kokoro Sensor, presenting a limitation in the interpretation. Previous research has primarily employed inter-group comparisons [17], highlighting the need for further investigation into reference values for emotional scores.

This study included patients with mild AD. As individuals with more severe dementia may show different responses, we would like to conduct further analysis with more participants spanning a wider range of disease stages. Furthermore, to

obtain more accurate measurement results, and to confirm the impact on treatment outcomes, future evaluations should consider multiple measurements, measurements over time, and environments without masks, as well as conditions that may enhance the accuracy of facial expression recognition.

## Conclusions

In this study, we conducted subjective mood evaluation and objective facial analysis using Kokoro Sensor during rehabilitation consisting of aerobic exercise, cognitive training, dual tasks (a combination of exercise and cognitive training), and creative activities in a group format with patients with MCI and dementia.

Patients with AD showed higher emotion scores for fear and surprise during tasks that were difficult and had a high risk of falling like dual tasks, and a lower emotion score for joy during tasks that used cognitive functions that were impaired like creative activities. At the same time, creative tasks that used multiple cognitive functions evoked joy in patients with MCI.

These emotion analysis findings can be used to choose tasks and adjust the difficulty level of a rehabilitation program for MCI and AD. We believe that this will facilitate the ability of patients with MCI and dementia to receive continuous preventive and therapeutic support.

## Supporting information

**S1 Fig. Four types of group rehabilitation.**
(PDF)

**S2 Fig. Patient's facial expression shooting scene.**
(PDF)

**S1 Data. Raw data 1–10.**
(ZIP)

**S2 Data. Raw data 11–20.**
(ZIP)

**S3 Data. Raw data 21–30.**
(ZIP)

## Acknowledgments

We would like to express our sincere gratitude to the MCI/AD patients and their family caregivers for their cooperation in this study, as well as to the physicians and rehabilitation staff who carry out their treatment on a daily basis.

## Author contributions

**Conceptualization:** Masaki Kamiya, Aiko Osawa, Izumi Kondo.

**Formal analysis:** Eri Otaka.

**Investigation:** Masaki Kamiya.

**Visualization:** Masaki Kamiya.

**Writing – original draft:** Masaki Kamiya.

**Writing – review & editing:** Aiko Osawa, Eri Otaka, Kenji Kato, Tatsuya Yoshimi, Hitoshi Kagaya, Izumi Kondo.

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
