## [Decision Letter · Decision Letter 0]

8 Oct 2024

PONE-D-24-14647Exploring emotion recognition in patients with mild cognitive impairment and Alzheimer’s dementia undergoing a rehabilitation program Emotion recognition in patients with dementiaPLOS ONE

Dear Dr. Kamiya,

Thank you for submitting your manuscript to PLOS ONE. After careful consideration, we feel that it has merit but does not fully meet PLOS ONE’s publication criteria as it currently stands. Therefore, we invite you to submit a revised version of the manuscript that addresses the points raised during the review process.

We look forward to receiving your revised manuscript.

Kind regards,

Jaiteg Singh

Academic Editor

PLOS ONE

Journal Requirements:

"NO authors have competing interests."

6. We note that Figures S1 and S2 in your submission contain copyrighted images. All PLOS content is published under the Creative Commons Attribution License (CC BY 4.0), which means that the manuscript, images, and Supporting Information files will be freely available online, and any third party is permitted to access, download, copy, distribute, and use these materials in any way, even commercially, with proper attribution. For more information, see our copyright guidelines: http://journals.plos.org/plosone/s/licenses-and-copyright.

a. You may seek permission from the original copyright holder of Figures S1 and S2 to publish the content specifically under the CC BY 4.0 license.

Reviewers' comments:

Reviewer's Responses to Questions

**Comments to the Author**

1. Is the manuscript technically sound, and do the data support the conclusions?

Reviewer #1: Yes

Reviewer #2: Yes

2. Has the statistical analysis been performed appropriately and rigorously?

Reviewer #1: Yes

Reviewer #2: Yes

3. Have the authors made all data underlying the findings in their manuscript fully available?

Reviewer #1: Yes

Reviewer #2: Yes

4. Is the manuscript presented in an intelligible fashion and written in standard English?

Reviewer #1: Yes

Reviewer #2: Yes

5. Review Comments to the Author

Reviewer #1: Strengths:

1. The introduction provides a clear and concise overview of the research gap related to the lack of evidence supporting the effects of group activities on people with dementia.

2. The statistical analyses appear to be appropriate for the data and research questions.

3. The authors have acknowledged the limitations of the study, including the impact of masks and the potential for underestimating smiles.

4. The analyzable percentage provides valuable information about the feasibility of facial analysis during group rehabilitation.

Improvements:

1. Discuss the potential clinical implications of the findings in more detail.

2. Expand the literature review to include more recent studies on the effectiveness of group activities for people with dementia with respect to facial analysis. (if any)

3. Investigate the impact of the rehabilitation program on emotion scores over time.

4. The authors could provide a more in-depth comparison of their findings with those of previous studies on facial analysis in dementia. This would help to contextualize their results and highlight the unique contributions of their research.

5. The authors include investigating the impact of facial analysis on treatment outcomes or exploring the feasibility of using facial analysis in different settings for future research.

6. Mention the citation where Kokoro senser has been used.

7. What measures are taken to calculate SD, how emotion scores are obtained through Kokoro sensor. Could you please elaborate at what scale the person is Happy, Sad, Anger etc.

8. A flowchart can be added to make the Assessment, Procedure and Analysis sections more easy to follow.

Reviewer #2: There should be some images or tables in introduction.

More latest papers needs to be discussed.

Conclusion should be specific

Add 2-3 more keywords

Describe the significance of each parameter of Table-1

6. PLOS authors have the option to publish the peer review history of their article (what does this mean? ). If published, this will include your full peer review and any attached files.

**Do you want your identity to be public for this peer review?** For information about this choice, including consent withdrawal, please see our Privacy Policy .

Reviewer #1: No

Reviewer #2: No

---

## [Author Response · Author response to Decision Letter 1]

4 Dec 2024

Dear Reviewer #1

Thank you very much for your review.

We read your review carefully and amended the paper on your pointed out.

Additional sentences and amendments are highlighted in yellow.

Comment 1

Discuss the potential clinical implications of the findings in more detail.

Answer 1

As you noted, explaining the clinical significance of this study's results is essential.

Patients with dementia commonly experience reduced spontaneity and language impairments, which can hinder their ability to express their thoughts accurately. This can lead to refusals of medical treatment or care, thus complicating the delivery of adequate medical attention. This study demonstrated that task preferences vary with dementia severity, and that subjective emotional expressions do not always align with objective facial expression analyses. These findings underscore the importance of tailoring assessments to specific contexts. Accurately capturing the emotions of dementia patients could also help to facilitate adjustments in task difficulty and selection of activities that engage them, enabling more personalized and respectful medical care.

Furthermore, the observed differences in task preferences between individuals with MCI and AD, linked to variations in cognitive function, further highlight the importance of tailoring interventions. By accurately interpreting facial expressions and adjusting tasks, medical professionals may be able to enhance patient motivation and promote sustained participation in preventive and therapeutic activities.

The above content has been incorporated into the text accordingly.

Line 334-339.

As such, the preferences for each task differ between patients with MCI and AD due to variations in cognitive function. Therefore, when presenting tasks to patients with MCI or AD, medical professionals can enhance motivation for continued prevention and treatment by accurately capturing each patient's facial expressions, and adjusting the tasks accordingly.

Line 369-378.

In particular, people with dementia may struggle to express their feelings appropriately, due to factors such as reduced spontaneity and language disorders, and may refuse medical treatment or care, making it difficult to provide adequate care. It has also been reported that one in two people with dementia do not voluntarily seek medical treatment [40]. By objectively evaluating facial expressions during interventions, along with asking the patient about the situation afterward, it may be possible to more accurately understand each patient's emotions. This could enable the provision of medical care that respects the individuality of each patient with dementia, such as adjusting the difficulty of interventions and selecting activities that interest them.

Comment 2

Expand the literature review to include more recent studies on the effectiveness of group activities for people with dementia with respect to facial analysis. (if any)

Answer 2

Thank you for pointing out the limited recent research on the usefulness of group activities for individuals with dementia. In response, we have now incorporated findings from a 2021 study on group activities utilizing robots and a 2023 study surveying medical professionals and family caregivers. The following content has been added to address this gap:

Line 55-59.

One report has also indicated that group activities involving robots for individuals with dementia can reduce nighttime awakenings and improve sleep quality [3]. Additionally, family members and medical professionals managing individuals with dementia have indicated that group activities are suitable for this population, provided that programming remains flexible [4]. Nonetheless, there is still inadequate evidence to support the effects of group activities on people with dementia [5, 6].

Comment 3

Investigate the impact of the rehabilitation program on emotion scores over time.

Answer 3

As you noted, it was necessary to measure emotions multiple times and to confirm changes over time. Additionally, as facial expressions were recorded during rehabilitation in standard medical practice, the four activity times varied slightly for each subject, making it difficult to accurately capture changes over time in a comprehensive manner. Moreover, as mentioned in the Limitation section, the analysis rate was low due to mask-wearing during the COVID-19 pandemic, which hindered the analysis of temporal changes. Nevertheless, confirming changes over time remains highly important. In the future, we aim to address the issues identified in this limitation and to conduct research under conditions that could improve the analysis rate, with multiple measures taken after the containment of COVID-19. A note regarding this future direction has been added to the Limitation section.

Line 402-406.

Furthermore, to obtain more accurate measurement results, and to confirm the impact on treatment outcomes, future evaluations should consider multiple measurements, measurements over time, and environments without masks, as well as conditions that may enhance the accuracy of facial expression recognition.

Comment 4

The authors could provide a more in-depth comparison of their findings with those of previous studies on facial analysis in dementia. This would help to contextualize their results and highlight the unique contributions of their research.

Answer 4

Thank you for your suggestion. We have cited an article on facial expression analysis for people with dementia during pain. We have already conducted such a study, but we have newly noted that this is the first time such a use as in this study has been done on people with dementia. Line 96-102.

Their findings revealed that facial expressions (e.g., frowns, nose wrinkles) and emotional expressions (e.g., disgust) influenced doctors' decisions, with aversion to ambiguity partially mediating therapeutic inertia. Additionally, prior studies have reported favorable inter-reliability outcomes when assessing pain in dementia patients using AI-based facial expression analysis [18]. However, no research utilizing facial expression analysis to optimize group activities for dementia patients has yet been conducted.

Comment 5

The authors include investigating the impact of facial analysis on treatment outcomes or exploring the feasibility of using facial analysis in different settings for future research.

Answer 5

Thank you for your suggestion. We would like to measure the results over time and verify if this is an outcome of the treatment effect. We would also like to verify whether facial expression analysis is useful in different environments and other situations. I have added an entry to Limitation as a future perspective. Line 402-406.

Furthermore, to obtain more accurate measurement results, and to confirm the impact on treatment outcomes, future evaluations should consider multiple measurements, measurements over time, and environments without masks, as well as conditions that may enhance the accuracy of facial expression recognition.

Comment 6

Mention the citation where Kokoro senser has been used.

Answer 6

As you pointed out, there were not enough citations regarding Kokoro Sensor. I added a study that investigated emotions when making treatment decisions in patients with multiple sclerosis. Line 92-98.

Saposnik et al. (2019) previously employed AFFDEX to explore the relationship between emotions, affective states (as interpreted through facial muscle activity and emotional expressions), and therapeutic inertia in physicians' treatment decisions for patients with multiple sclerosis [17]. Their findings revealed that facial expressions (e.g., frowns, nose wrinkles) and emotional expressions (e.g., disgust) influenced doctors' decisions, with aversion to ambiguity partially mediating therapeutic inertia.

Comment 7

What measures are taken to calculate SD, how emotion scores are obtained through Kokoro sensor. Could you please elaborate at what scale the person is Happy, Sad, Anger etc.

Answer 7

As noted, we implemented the following measures:

Times when the neutral value was 50 or higher, including conversations unrelated to rehabilitation, breaks, and waiting periods, were excluded from the analysis, as they fell outside of the study's scope.

The average emotion value for each subject, extracted through this process, was used as the representative value, while variation among subjects was calculated as the standard deviation (SD).

The Kokoro Sensor's emotion score estimates emotions using AI technology specialized in facial expression recognition. However, the calculation algorithm and technical details regarding specific expressions and emotions have not been publicly disclosed. The general facial expression scoring process is as follows:

a. Face detection: Identifies the face area within the video or image.

b. Feature point extraction: Extracts facial feature points, such as the eyes, nose, and mouth from the detected face.

c. Facial expression recognition: Identifies specific facial expressions based on the extracted feature points.

d. Facial expression scoring: Assigns a score for each facial expression based on its intensity, expressed as a number from 0 to 100.

e. Emotion scoring: Links specific facial expressions, as defined by the Facial Action Coding System (FACS), to corresponding emotions and calculates an emotion score from the facial expression value, expressed as a number from 0 to 100.

For the degree of happiness, sadness, anger, etc., previous studies have typically focused on inter-group comparisons of mean values, without establishing quantitative reference standards. Thus, our study similarly relies on inter-group comparisons, and the absence of standardized baseline values has been added as a limitation. Line395-399.

In this study, no established reference values exist for interpreting the emotional scores calculated by Kokoro Sensor, presenting a limitation in the interpretation. Previous research has primarily employed inter-group comparisons [17], highlighting the need for further investigation into reference values for emotional scores.

Comment 8

A flowchart can be added to make the Assessment, Procedure and Analysis sections more easy to follow.

Answer 8

As you pointed out, it was difficult to understand the flow of Method. We have created a flowchart as follows.

Dear Reviewer #2

Thank you very much for your review.

We read your review carefully and amended the paper on your pointed out.

Additional sentences and amendments are highlighted in yellow.

Comment 1

There should be some images or tables in introduction.

Answer 1

As you pointed out, it was difficult to understand the process of facial expression analysis by AI for people with MCI and dementia and how we estimated that facial expression analysis could be used for them. Therefore, we inserted Fig 1. in the introduction.

Comment 2

More latest papers needs to be discussed.

Answer 2

As you pointed out, we did not address the newer papers.

I have added the papers from around 2020 in the following section.

A paper on the effectiveness of group activities for dementia. Line 55-59.

・Effectiveness of group activities using robots

Jøranson N, Olsen C, Calogiuri G, Ihlebæk C, Pedersen I. Effects on sleep from group activity with a robotic seal for nursing home residents with dementia: a cluster randomized controlled trial. Int Psychogeriatr. 2021; 33(10): 1045-1056. pmid: 32985396.

・The usefulness of group activities from the perspective of families and health care providers

O'Rourke HM, Jeffery N, Walsh B, Quark S, Sidani S. Understanding Acceptability of Group Leisure Activities Used to Address Loneliness Among People Living With Dementia: An Exploratory Mixed-Methods Study. Can J Aging. 2023; 42(4): 565-575. pmid: 37492945.

・Review of Dementia Interventions

Xue D, Li PWC, Yu DSF, Lin RSY. Combined exercise and cognitive interventions for adults with mild cognitive impairment and dementia: A systematic review and network meta-analysis. Int J Nurs Stud. 2023; 147: 104592. pmid: 37769394.

・Prior literature on Kokoro Sensor used in this study. Line 92-96.

Saposnik G, Oh J, Terzaghi MA, Kostyrko P, Bakdache F, Montoya A, Jaja BNR, Nisenbaum R, Ruff CC, Tobler PN. Emotional expressions associated with therapeutic inertia in multiple sclerosis care. Mult Scler Relat Disord. 2019; 34: 17-28. pmid: 31226545.

The AFFDEX used in this paper is a similar application to Kokoro Sensor.

Papers on Facial Expression Analysis

・Mutual reliability of pain assessment and AI-based facial expression analysis in people with dementia. Line98-100.

Atee M, Hoti K, Parsons R, Hughes JD. A novel pain assessment tool incorporating automated facial analysis: interrater reliability in advanced dementia. Clin Interv Aging. 2018; 13: 1245-1258. pmid: 30038491.

Paper on decision-making in people with dementia. Line 369-373.

Heale R. Dementia care and treatment issues. Evid Based Nurs. 2020; 23(2): 40-42. pmid: 32209613.

Comment 3

Conclusion should be specific

Answer3

As you pointed out, the conclusion was abstract, so we made it concrete.

Line 409-422.

Conclusions

In this study, we conducted subjective mood evaluation and objective facial analysis using Kokoro Sensor during rehabilitation consisting of aerobic exercise, cognitive training, dual tasks (a combination of exercise and cognitive training), and creative activities in a group format with patients with MCI and dementia.

Patients with AD showed higher emotion scores for fear and surprise during tasks that were difficult and had a high risk of falling like dual tasks, and a lower emotion score for joy during tasks that used cognitive functions that were impaired like creative activities. At the same time, creative tasks that used multiple cognitive functions evoked joy in patients with MCI.

These emotion analysis findings can be used to choose tasks and adjust the difficulty level of a rehabilitation program non-pharmacological therapies for MCI and AD. We believe that this will facilitate the ability of patients with MCI and dementia to receive continuous preventive and therapeutic support.

Comment 4

Add 2-3 more keywords

Answer4

As you indicated, we have added two to the five key words that we submitted.

Key words: rehabilitation, mild cognitive impairment, Alzheimer’s disease, facial analysis, artificial intelligence, emotion, activity

Comment 5

Describe the significance of each parameter of Table-1

Answer 5

As you pointed out, there was no description of the importance of each parameter.

We have added the following description and importance of the parameters in the Assessment section of the text. Line 159-169.

Depression is considered a risk factor or precursor for dementia [23], while its presence should be assessed as it may influence both facial expressions and subjective well-being. For evaluation of cognitive function, the Mini-Mental State Examination-Japanese (MMSE-J) [24] was used. We further used the Frontal Assessment Battery (FAB) [25], which is associated with dementia assessment scales and can be used to assess frontal lobe dysfunction, as well as Raven's Progressive Matrices (RCPM) [26], which evaluates visuospatial cognitive function. The Neuropsychiatric Inventory (NPI) [27] was used to evaluate behavioral and psychological symptoms of dementia (BPSD). Further, in order to indicate the daily activity level, activities of daily living (ADL) were evaluated using the Barthel Index (BI) [28] and the Frenchay Activities Index (FAI) [29].

---

## [Decision Letter · Decision Letter 1]

18 Mar 2025

Exploring emotion recognition in patients with mild cognitive impairment and Alzheimer’s dementia undergoing a rehabilitation program Emotion recognition in patients with dementia

PONE-D-24-14647R1

Dear Dr. Masaki Kamiya,

We’re pleased to inform you that your manuscript has been judged scientifically suitable for publication and will be formally accepted for publication once it meets all outstanding technical requirements.

Kind regards,

Miray Budak

Academic Editor

PLOS ONE

Reviewer's Responses to Questions

**Comments to the Author**

1. If the authors have adequately addressed your comments raised in a previous round of review and you feel that this manuscript is now acceptable for publication, you may indicate that here to bypass the “Comments to the Author” section, enter your conflict of interest statement in the “Confidential to Editor” section, and submit your "Accept" recommendation.

Reviewer #1: All comments have been addressed

Reviewer #3: All comments have been addressed

2. Is the manuscript technically sound, and do the data support the conclusions?

Reviewer #1: Yes

Reviewer #3: Yes

3. Has the statistical analysis been performed appropriately and rigorously?

Reviewer #1: Yes

Reviewer #3: Yes

4. Have the authors made all data underlying the findings in their manuscript fully available?

Reviewer #1: Yes

Reviewer #3: Yes

5. Is the manuscript presented in an intelligible fashion and written in standard English?

Reviewer #1: Yes

Reviewer #3: Yes

6. Review Comments to the Author

Reviewer #1: (No Response)

Reviewer #3: All reviewers' concerns have been properly addressed. I recommend the manuscript for publication on PLOS One.

7. PLOS authors have the option to publish the peer review history of their article (what does this mean? ). If published, this will include your full peer review and any attached files.

**Do you want your identity to be public for this peer review?** For information about this choice, including consent withdrawal, please see our Privacy Policy .

Reviewer #1: No

Reviewer #3: No

---

## [Editor Report · Acceptance letter]

PONE-D-24-14647R1

PLOS ONE

Dear Dr. Kamiya,

I'm pleased to inform you that your manuscript has been deemed suitable for publication in PLOS ONE. Congratulations! Your manuscript is now being handed over to our production team.

Kind regards,

on behalf of

Dr. Miray Budak

Academic Editor

PLOS ONE
